# Bone Marrow-Derived Mesenchymal Stromal Cells (MSCs) Modulate the Inflammatory Character of Alveolar Macrophages from Sarcoidosis Patients

**DOI:** 10.3390/jcm9010278

**Published:** 2020-01-19

**Authors:** Ian McClain Caldwell, Christopher Hogden, Krisztian Nemeth, Michael Boyajian, Miklos Krepuska, Gergely Szombath, Sandra MacDonald, Mehrnoosh Abshari, Joel Moss, Lynn Vitale-Cross, Joseph R Fontana, Eva Mezey

**Affiliations:** 1Adult Stem Cell Section, National Institute of Dental and Craniofacial Research (NIDCR), National Institutes of Health (NIH), Bethesda, MD 20892, USA; mcclainian@gmail.com (I.M.C.); christopher.hogden@nih.gov (C.H.); michael_boyajian@brown.edu (M.B.); krepuskam@hotmail.com (M.K.); lcross@dir.nidcr.nih.gov (L.V.-C.); mezeye@nih.gov (E.M.); 2National Heart, Lung, and Blood Institute (NHLBI), NIH, Bethesda, MD 20892, USA; sm166d@nih.gov (S.M.); mossj@nhlbi.nih.gov (J.M.); jf345d@nih.gov (J.R.F.); 3Combined Technical Research Core, National Institute of Dental and Craniofacial Research, National Institutes of Health, Bethesda, MD 20892, USA; absharim@nidcr.nih.gov; 4Stem Cell Laboratory, Department of Dermatology, Venerology and Dermato-oncology, Semmelweis University, Budapest 1085, Hungary; szomger@gmail.com

**Keywords:** sarcoidosis, alveolar macrophages, bone marrow stromal cells, cell therapy

## Abstract

Sarcoidosis is a devastating inflammatory disease affecting many organs, especially the lungs and lymph nodes. Bone marrow-derived mesenchymal stromal cells (MSCs) can “reprogram” various types of macrophages towards an anti-inflammatory phenotype. We wanted to determine whether alveolar macrophages from sarcoidosis subjects behave similarly by mounting an anti-inflammatory response when co-cultured with MSCs. Fifteen sarcoidosis and eight control subjects underwent bronchoscopy and bronchoalveolar lavage (BAL). Unselected BAL cells (70–94% macrophages) were isolated and cultured with and without MSCs from healthy adults. Following stimulation of the cultured cells with lipopolysaccharide, the medium was removed to measure interleukin 10 and tumor necrosis factor alpha (IL-10 and TNF-α). In two additional sarcoidosis subjects, flow cytometry was used to study intracellular cytokines and surface markers associated with alveolar macrophages to confirm the results. Unselected BAL cells from sarcoidosis subjects co-cultured with MSCs showed a reduction in TNF-α (pro-inflammatory M1) and an increase in IL-10 (anti-inflammatory M2) in 9 of 11 samples studied. Control subject samples showed few, if any, differences in cytokine production. Unselected BAL cells from two additional patients analyzed by flow cytometry confirmed a switch towards an anti-inflammatory state (i.e., M1 to M2) after co-culture with MSCs. These results suggest that, similarly to other macrophages, alveolar macrophages also respond to MSC contacts by changing towards an anti-inflammatory phenotype. Based on our results, we hypothesize that mesenchymal stromal cells applied to the airways might alleviate lung inflammation and decrease steroid need in patients with sarcoidosis.

## 1. Introduction

Bone marrow stromal cells (called MSCs) regenerate skeletal cells in the marrow niche environment and support hematopoiesis. MSCs are a very heterogeneous population of cells [1] that can differentiate to various lineages of mesenchymal tissues, including bone, muscle, and adipose tissue [2]. The observation that they can modulate the immune system resulted in extensive research on their use to treat inflammatory conditions, including T-cell-mediated diseases such as graft-versus-host disease (GVHD) [3], as well as granulomatous disorders such as Crohn’s disease [4]. MSCs decrease inflammation and promote tissue repair through cell–cell interactions including mitochondrial transfer [5], and by way of paracrine effects [6,7]. In clinical trials to date, they have demonstrated an excellent safety profile [8,9].

It is generally agreed that MSCs can reprogram activated macrophages to adopt an anti-inflammatory state (i.e., the phenotype moves from M1 towards M2) [10,11,12,13]. Cyclooxygenase-2 (COX-2) signaling contributes to driving this change and was first shown in vivo in a mouse model of sepsis [14]. Inflammatory stimuli, such as lipopolysaccharide (LPS) and TNF-α, interact with toll-like receptors in MSCs to induce nuclear factor kappa B (NF-kB), which in turn increases the production of COX-2. This enzyme increases the production of Prostaglandin E2 (PGE2) [14], which interacts with receptors on macrophages to increase IL-10 and decrease TNF-α production. MSCs from a variety of tissues seem to behave the same way [13]. Engulfment of apoptotic MSCs by macrophages was recently shown to increase the production of other immune suppressive factors, demonstrating an additional mechanism of action [15].

Sarcoidosis is a multisystem granulomatous inflammatory disease propelled primarily by dysregulated macrophage TNF-α production and increased T-helper-1 (Th1 and Th17) cell proliferation [16]. It often involves the lungs and the lymphatic system, but the etiology remains largely unknown [17,18]. Oligoclonal expansion of patient T-cells suggests that sarcoidosis may be an antigen-driven immune process [19,20] in genetically disposed hosts [21,22]. Macrophage activation plays a key role in disease pathogenesis [23], and macrophage-directed recruitment of T-helper cells is now recognized as having significant importance in the disease [24]. Currently available treatment options include glucocorticoids, cytotoxic drugs, and anti-TNF-α monoclonal antibodies, most of which have high risk/benefit profiles [25].

We hypothesized that human MSCs may have a beneficial effect if they can reprogram alveolar macrophages (AMs) of patients with sarcoidosis. To look for such an effect in vitro, we recruited biopsy-confirmed sarcoidosis subjects and control subjects to provide bronchoalveolar lavage (BAL) samples for co-culture experiments with human MSCs. 

It has been shown that pro-inflammatory macrophages from a variety of sources convert to anti-inflammatory phenotype when co-cultured with MSCs. The results presented here demonstrate for the first time that AMs from patients with sarcoidosis behave similarly. The conversion is achieved by MSCs inducing IL-10 and decreasing TNF-α production of AMs. Given their known safety profile, we suggest that MSCs may offer a safe alternative or adjunctive treatment to drug therapy.

## 2. Methods

### 2.1. Clinical Procedure 

Seventeen adult sarcoidosis subjects (Scadding [26] stages 0–IV), who were receiving minimal or no anti-inflammatory treatment, and eight control subjects were recruited to the NIH and consented to protocol 96-H-0100 (NCT00001532) to undergo bronchoscopy with BAL. Subjects received a medical evaluation and underwent laboratory and pulmonary function testing prior to the procedure. They received IV conscious sedation and topical anesthesia of their airways. Lavages were performed with 270–300 mL of sterile saline in aliquots of 30 mL each; the right and left upper lobes were primarily targeted. Samples were aspirated into Lukens traps and placed on ice. Approximately 10 ml of the BAL fluid return (BALF) was analyzed for viability and via cytometry to obtain white blood cell (WBC) counts. Viability was determined using Trypan Blue staining, while cell differential counts were performed after cytocentrifuge and Diff-Quik^TM^ staining. The remaining volume was submitted for in vitro co-culture experiments with MSCs. For details of culture and co-culture conditions, as well as the ELISA protocol, please see the online data supplement.

### 2.2. Cell Culture

Cryopreserved clinical-grade human MSCs derived from iliac crest biopsies of healthy adult donors were obtained from the Clinical Center of the NIH, Bethesda, MD under protocol 10-CC-0053 (NCT01071577), expanded, and cryopreserved in aliquots of 1 to 4 million cells. For testing, the aliquots were removed from liquid nitrogen storage, thawed, and cultured in minimal essential medium (MEM-α) supplemented with 10% fetal bovine serum (FBS), 1% GlutaMax, and 1% PenStrep (herein named “MSC medium”). Macrophage medium consisted of RPMI 1640 supplemented with 10% FBS, 1% PenStrep and 0.00035% beta-mercaptoethanol. Freezing medium consisted of 50% MSC or macrophage medium, depending on the cell type, 40% FBS, and 10% dimethyl sulfoxide (DMSO). Cells were expanded in culture in 5% CO_2_ and 20% O_2_ at 37 °C. 

Unselected BAL cells, with 70–94% AMs, were filtered upon receipt from the clinic in preparation for the co-culture assay. BAL cells were centrifuged, re-suspended in macrophage medium, counted, and cultured overnight at 37 °C in a 96-well plate with and without passage 4–6 clinical-grade MSCs. Most samples were plated in a ratio of 100,000 BAL cells to 10,000 MSCs in 200 µL of macrophage medium per well in eight replicates. One-half of the replicates were stimulated with 1 µg/mL of LPS the following morning, and supernatants were removed at 6 and 24 h for ELISAs to determine concentrations of TNF-α and IL-10, respectively.

Passage 5 clinical-grade MSCs from two healthy adult donors were combined and plated into Nunc™ 6-well UpCell™ Surface culture plates (ThermoFisher Scientific, Waltham, MA, USA) at a concentration of 3–4 × 10^4^ cells per well in MSC medium. Cells were incubated overnight at 37 °C with 5% CO_2_ and 20% O_2_. MSC medium was removed and replaced with 3 ml of macrophage medium with and without 3–4 × 10^5^ unselected BAL cells. Additionally, BAL cells were plated in the absence of MSCs. Plates were incubated for 16 hours as above.

### 2.3. Osteogenic and Adipogenic Differentiation Assay of MSCs

To demonstrate stem cell properties of MSCs, in vitro bone and adipose differentiation assays were performed as previously reported [27]. In brief, MSCs were plated in 6-well plates in complete medium and were cultured either in osteogenic or adipogenic medium for 16 days, at which time cells were fixed and stained with Alizarin Red or Oil Red, respectively. Representative microscopic pictures are shown (Appendix A). 

### 2.4. ELISA 

Pilot experiments were performed to determine the optimal ratio of MSCs to mononuclear BAL cells and time points for cytokine detection. To harvest the samples, co-culture plates were centrifuged, and the supernatants were transferred to low-absorbance plates for temporary storage at −20 °C or immediate use in an ELISA. A 1:5 dilution of the supernatant was performed to prevent the TNF-α concentrations from exceeding the concentration of the highest standard (1000 pg/mL).

ELISAs for human IL-10 and TNF-α were performed using DuoSet ELISA kits (DY217B, DY210, DY417, DY410) from R&D Systems (Minneapolis, MN, USA) according to the manufacturer’s instructions. The plates were analyzed using a Turner BioSystems (Sunnyvale, CA, USA) Modulus Microplate Reader calibrated for ELISA analysis at 450 nm using 3,3’,5,5’-Tetramethylbenzidine (TMB). ELISAs were repeated for samples that markedly overshot the high standard or had inexplicably high variance among replicates. 

### 2.5. Flow Cytometry

Antibodies used were anti-CD68 Allophycocyanin (APC), anti-CD11c Phycoerythrin (PE), anti-CD11c Peridinin Chlorophyll Protein Complex (PerCP)-Cy5.5, anti-CD80 PE-Cy5, anti-CD163 Brilliant Violet 711, anti-CD206 APC/Fire 750, anti-TNF-α Brilliant Violet 785, and anti-IL-10 PE (BioLegend, San Diego, CA, USA). 

For surface staining, cells were harvested by placing plates at 4 °C for 30 min and trituration to avoid the potential loss of select M2 cell surface markers by enzymatic degradation. Fc receptors were blocked with Human TruStain FcX (BioLegend) for 10 min at room temperature. Surface antibodies were added and incubated for 20 min at 4 °C in the dark and then washed with 5% FBS in PBS three times. Cell viability was determined using 4’,6-diamidino-2-phenylindole (DAPI). Single-stain compensation controls were prepared using UltraComp eBeads (Invitrogen, Waltham, MA, USA) according to the manufacturer’s instructions. Cell surface staining was analyzed using a BD LSRFortessa (Becton, Dickinson and Company, BD Biosciences, San Jose, CA, USA) immediately following staining. FlowJo software version 10.5.3 (FlowJo LLC, Ashland, Oregon) was used to analyze the acquired data.

For detection of cytokine expression inside the AMs, Monensin (eBioscience, Waltham, MA, USA) and Brefeldin A (eBioscience) were titrated to determine the protein transport inhibitor best suited for our cytokines and cells of interest [11]. Monensin was chosen and added at a final concentration of 2 M 4 h prior to harvesting cells to block secretion of cytokines. Cells were harvested as indicated above. Briefly, after staining with Zombie UV™ Fixable Viability Kit (BioLegend) and blocking of Fc receptor-mediated nonspecific binding, cells were stained with antibodies against surface markers for 20 min at 4 °C in the dark and washed as above. Cells were fixed for 20 min at room temperature and permeabilized using the Intracellular Fixation and Permeabilization Buffer Set (eBioscience) according to the manufacturer’s instructions. Cytokine-specific antibodies were added and incubated for 30 min at 4 °C in the dark and washed. Single-stain compensation controls were prepared as above. The Zombie UV™ single-stain compensation control was prepared using ArC™ Amine Reactive Compensation Bead Kit (Invitrogen) according to the manufacturer’s instructions. Samples were analyzed for flow cytometry as above. For all gatings, fluorescence minus one controls (FMOs) were used.

## 3. Results

The magnitude of any change in cytokine production was found to be positively correlated with the number of MSCs in the co-culture (Appendix A). To produce consistent, statistically significant changes in cytokine production, a ratio of 1 MSC to 10 monocyte/macrophage/BAL cells was sufficient. After the conclusion of the pilot experiments, this ratio was used throughout the study. 

Seventeen subjects with biopsy-confirmed sarcoidosis and a compatible clinical history were selected to participate in the study. They were using little or no anti-inflammatory drugs at the time of the BAL harvest (Table 1). All five Scadding stages were represented in the sarcoidosis cohort: Stage 0: 3 (17.6%); Stage I: 3 (17.6%); Stage II: 1 (5.9%); Stage III: 6 (35.3%); and Stage IV: 4 (23.5%). Eight healthy adults were recruited as controls. Sarcoidosis subjects had a significantly lower mean forced vital capacity (FVC), forced expired volume in 1 second (FEV_1_), and diffusion adjusted for hemoglobin (DLCO adj.) (Appendix A); and significantly higher serum angiotensin-converting enzyme (ACE) levels and peripheral blood monocyte numbers compared with control subjects (Appendix A). Peripheral blood lymphocytes were similar in the two cohorts (Appendix A); however, sarcoidosis subjects had a significantly greater percentage of lymphocytes in their BAL (Table 2). 

ELISA assays were performed to detect the cytokine (TNF-α and IL-10) production by the isolated BAL cells in culture with and without human MSCs. Samples of medium were collected 6 and 24 h after LPS stimulation of co-cultured MSCs and BAL cells from sarcoidosis and control subjects. ELISA measurements of these co-culture supernatants showed that BAL cells from sarcoidosis subjects significantly decreased their TNF-α production (*p* = 0.029) and increased their IL-10 production (*p* = 0.011), unlike cells from control subjects (Figure 1A,B). This result seems to reflect a shift from a pro-inflammatory M1 to a more anti-inflammatory M2 phenotype. 

Cytokine production was also evaluated in co-cultures not stimulated with LPS. In these samples, IL-10 and TNF-α were either not detected or barely detectable in the culture medium. MSCs alone did not produce any measurable IL-10 or TNF-α in the presence or absence of LPS, indicating that BAL cells produced the measured cytokines. 

To confirm this and determine whether it was macrophages in the BAL preparations that made the cytokines we measured, we studied intracellular IL-10 and TNF-α in a population of AMs identified by flow cytometry and also examined macrophage-associated cell surface antigens. Because the detection of the cytokines was quite sensitive in the analyzed cells, it was unnecessary to add LPS to stimulate their production. BAL cells from two additional sarcoidosis subjects were studied. Subject 16 was more symptomatic and functionally impaired, and appeared to have a more active disease process (Appendix A). Subject 17 had clinically inactive disease, a lower number of AMs, and a lower peripheral CD4^+^/CD8^+^ ratio (Appendix A). Sixteen hours after BAL cells were placed in culture with and without MSCs, they were harvested, processed, and analyzed by flow cytometry. We focused on CD206+ cells, which are considered to be the AMs [28,29] among the BAL cells. In AMs from subject 16, twice as many cells produced IL-10 when they were co-cultured with MSCs (22.1%) as they did when cultured without them (11.6%) (Figure 2A,B). In parallel with the increase in IL-10, there was a decrease in TNF-α (from 2.27% to 1.07%) (Figure 2C,D), resulting in a 4-fold increase in the IL-10/TNF-α ratio (Figure 3). AMs from subject 17 seemed less MSC-sensitive. While the IL-10 level more than doubled (3.05% to 8.67%) (Fig. 4A and 4B), there was a slight increase in TNF-α (0.94% to 1.25%) (Figure 4C,D), and the IL-10/TNF-α ratio only increased 2-fold (Figure 3B).

Consistent with recent publications [28,29], our data confirm that CD206 was nearly ubiquitously expressed in the AΜ population of the sarcoidosis subjects investigated (Figure 2 and Figure 4). In both subjects, the mean fluorescent intensity of CD206 increased upon co-culture with MSCs (Figure 2, Figure 3A and Figure 4). CD163, a M2 macrophage marker, increased in AMs co-cultured with MSCs in subject 16 (10.9% to 15.8%) (Figure 2E,F), but decreased in subject 17 (27.4% to 24%) (Figure 4E,F) compared with AMs cultured alone. Furthermore, there was no CD163 heterogeneity observed in the AMs (Figure 2E,F and Figure 4E,F).

At the time point studied, CD11c and CD80 (two M1 macrophage markers) in AMs co-cultured with MSCs increased from 25.3% to 40.1% and from 20.9% to 41.1%, respectively, in subject 16 (Appendix A). In subject 17, CD11c and CD80 decreased from 56.9% to 55.2% and from 49.8% to 41.9%, respectively, compared with AMs alone (Appendix A). Similar results for the surface markers were seen in fixed and unfixed cells.

## 4. Discussion

Our results demonstrate that alveolar macrophages from an inflammatory environment (sarcoidosis patients) respond to cues from co-cultured MSCs and change to a more anti-inflammatory phenotype. Although this effect has been reported in other tissue macrophages, the finding that AMs from patients with sarcoidosis respond to MSCs has clinical significance and suggests that a cellular therapy should be tested. 

We have shown previously that the inflammatory environment has a profound effect on MSC-mediated immunomodulation. Extracellular inflammatory signals, including pro-inflammatory cytokines, and small molecules such as prostaglandins and nitric oxide are able to augment the production of MSC-derived immunomodulatory factors and ultimately lead to enhanced immunosuppression [14]. Alveolar macrophages isolated from sarcoidosis patients are in an activated state and produce significantly higher levels of inflammatory cytokines, such as TNF-α compared with control AMs [30]. When MSCs are exposed to this sarcoidosis-derived inflammatory milieu (as opposed to a homeostatic, control macrophage-produced environment) they are more likely to induce an anti-inflammatory character in AMs, which could ultimately lead to a more consistent and substantial reduction in macrophage-secreted TNF-α and an increase in IL-10 levels.

Bone marrow stromal cells have been shown to have multiple immunomodulatory properties, including reprogramming of macrophages. MSCs change macrophages functionally and phenotypically from activated, pro-inflammatory (M1) cells to anti-inflammatory (M2) cells [14]. This reprogramming is mediated by the COX-2 pathway in MSCs, which increases their PGE2 production. PGE2 then binds to the E2 and E4 receptors on the surface of macrophages to induce increased IL-10 production. In addition, MSCs can attenuate T-cell activation and induce tolerance by interfering with the maturation of dendritic cells [31]; this may be due in part to induction of Signal Transducer and Activators of Transcription-3 (STAT-3) signaling and cell-to-cell contact [32]. Given the above, we questioned if MSCs might alleviate lung inflammation in patients with sarcoidosis. 

Sarcoidosis is a multi-system disease characterized by noncaseating granulomatous inflammation in a Th1/Th17 cytokine environment with elevated levels of IL-2, IL-12, IL-17, TNF- α, and IFN-γ [24]. Potential targets for therapy include TNF-α-blocking agents (monoclonal antibodies) and other agents that target pro-inflammatory pathways [33]. In a small phase I trial, four patients with chronic sarcoidosis, stages II and III, were given placental mesenchymal-like cells intravenously twice in a one-week period and followed for two years. Two of the four patients were able to discontinue prednisone and had significant improvements in their chest radiographs [34]. 

Due to the lack of an appropriate animal model, we tested this hypothesis by isolating primary cells from subjects’ BAL and cultured briefly in the presence and absence of healthy human MSCs. This “ex-vivo” experimental setup mimics the interaction between MSCs and AMs. We looked for changes in cytokine production of the isolated AMs as well as changes in surface marker expression [35].

While changes in AM IL-10 and TNF-α production in the presence of MSCs (Figure 1B and Figure 3B) indicated transition to an anti-inflammatory state, surface marker expression of the AMs was not typical of what is seen in a M2 shift (Appendix A). This might be a result of different timing of expression between surface markers and cytokines. However, recent evidence suggests that AMs exhibit a hybrid M1/M2 phenotype [28,29], implying that we do not understand the unique dynamics of surface marker expression in this specific macrophage population at the current time. Therefore, the cytokine data may provide a more reliable readout of the inflammatory state of the AM population, and we may need to reconsider the M1/M2 paradigm as a general feature of all macrophages [36]. 

In our cytokine ELISA assays of co-culture medium, two sarcoidosis subjects did not follow the trend of decrease in TNF-α and increase in IL-10 (like all the other patients). In one subject (19 in Figure 1), both TNF-α and IL-10 decreased in the culture medium when his/her BAL cells were co-cultured with MSCs in the presence of LPS. This patient had an active sinus infection at the time of BAL harvest that might account for the deviation. The other unusual response had a slight increase in TNF-α, but a larger increase in IL-10 production (25 in Figure 1). This patient failed to stop his/her steroid intake as all the others did. It is important to mention though, that in spite of the deviation from the other responses, even in these two patients the change induced in the AMs by the MSCs was tilted towards anti-inflammatory change, shifting the ratio towards less TNF-α and more IL-10. In a separate assay, one of the two sarcoidosis subjects whose BAL was analyzed by flow cytometry showed a similar change—a small increase in TNF-α and a larger increase in IL-10. This patient had clinically inactive disease overall. It seems that macrophage-derived pro-inflammatory signals are necessary for the MSCs to exert their effect efficiently [37].

The use of MSCs in humans with pulmonary diseases has recently been reviewed [38,39]. We hypothesized that MSC cellular therapy, as a local treatment through the airways, might decrease lung inflammation in patients with sarcoidosis. The route of delivery of MSCs may also influence treatment efficacy and can affect which molecular pathways lead to therapeutic immunomodulation. When treating an airway disease, such as sarcoidosis, MSCs can be injected either intravenously or delivered directly into the airways. In rodents, intravenously injected MSCs are entrapped in the pulmonary vasculature, and hence will be concentrated in the lungs [40]. In humans, MSCs can be found in the lungs only temporarily after intravenous infusion, which is followed by quick redistribution into the reticuloendothelial system, primarily the spleen and the liver [41]. Intravascular delivery will initiate a process called instant-blood-mediated inflammatory reaction (IBMIR), which is a complex inflammatory phenomenon that will eventually eliminate living/apoptotic MSCs from capillaries and may result in a more sustained monocyte/macrophage-mediated anti-inflammatory state [42,43]. Intrabronchial delivery of MSCs will avoid IBMIR and will result in direct interactions with alveolar inflammatory cells, including alveolar macrophages and lymphocytes. Although it is unclear if either of these delivery methods would be superior in terms of treatment efficacy, intrabronchial delivery of MSCs may lead to a more concentrated intrapulmonary effect in humans, thereby increasing treatment efficacy, needing a lower number of cells and minimizing the risks of possible systemic, IBMIR-related thrombo-inflammatory side effects [42].

Our data suggest that MSCs should be further studied as a potential therapy for pulmonary sarcoidosis due to their ability to decrease inflammation and possibly reduce the need for steroids. MSCs have been used in clinical settings for over a decade now, and although they were not always efficacious (depending on the disease and trial conditions), the consensus is that the therapy is safe [9,44,45]. A pilot study in pre-term infants suggested that intratracheal administration of MSCs was feasible and free from significant adverse events [46]. Endobronchial delivery of MSCs is being tested in patients with idiopathic pulmonary fibrosis (NCT01919827); the results are pending. Phase 1 clinical trials using MSCs from bone marrow [47], placenta [48], and adipose tissue [49] in the treatment of idiopathic pulmonary fibrosis have demonstrated a safe profile without evidence of worsening fibrosis due to the therapy. Although idiopathic pulmonary fibrosis and sarcoidosis are different diseases, these data are important when thinking about cell therapy of sarcoidosis as M2 macrophages have been suggested to increase pulmonary fibrosis [50]. Whether MSCs inhibit or promote fibrosis in the human lung is currently unknown, although there is evidence in mice of both pro and antifibrotic activity [51].

Interestingly, in mice, the ontogeny of the macrophages appears to be crucial for the development of fibrosis [52]. Therefore, targeting specific macrophage populations to alter their profibrotic profile or prevent their entrance into the lung may prove beneficial. Unfortunately, there is currently no available mouse model for sarcoidosis, so the applicability of these studies to our patient population is unknown. 

A variety of strategies have been tested to improve the efficiency of MSC therapy in respiratory diseases [53]. Kardia et al. [54] used fibroblast cells from rabbits, which resemble MSCs, to show that cells grown in tissue culture can be aerosolized without showing any sign of stress (i.e., vacuolation) afterwards. Most of the fibroblasts survived and proliferated rapidly. A recent study reported that MSC viability was better via compressed nebulization compared with ultrasonic or mesh device delivery [55]. In addition to optimizing the mode of administration, other details such as dose and timing need to be investigated. One advantage of MSCs is that they can be banked, and frozen aliquots can be studied and used. While questions have been raised about a possible loss of efficacy when cells are frozen [56], in certain clinical scenarios this may not be a serious problem [57]. Because of the possibility that cryopreservation may compromise the immunomodulatory activity of MSCs however [45], we used cultured, freshly trypsinized cells in our studies. It is unclear if, and to what extent, freezing would change the effect of MSCs on alveolar macrophages. Only future clinical trials can tell if there are clinically relevant differences between these two cellular products.

Given the known safety of MSC cellular therapy and the efficacy of these cells in countering inflammation, we suggest that their use in sarcoidosis could be steroid-sparing or supplant the need for systemic steroids. 

## Figures and Tables

**Figure 1 jcm-09-00278-f001:**
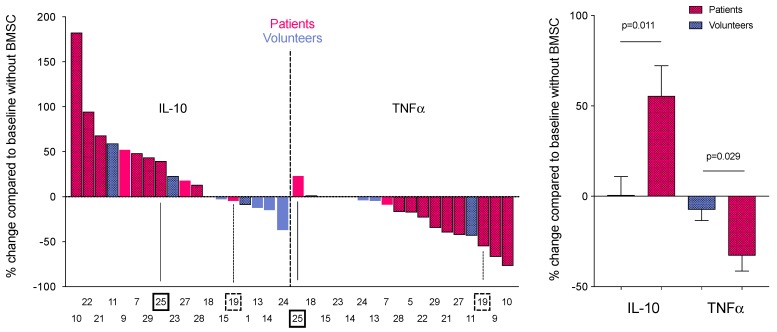
Percent change in cytokine production of bone marrow stromal cell (MSC) and bronchoalveolar lavage (BAL) cell co-cultures stimulated with lipopolysaccharide (LPS). (**A**) Cytokine (IL-10—left half and TNF-α—right half) concentrations were measured from tissue culture medium of LPS-stimulated co-cultures of MSCs and BAL cells from sarcoidosis (red) and control (blue) subjects. Of the 11 cases presented, 2 sarcoidosis samples (19 and 25) diverged from the trend in one cytokine or the other (indicated by arrows and asterisks). Control subject samples showed few, if any, differences in cytokine production with no obvious trend. (**B**) Average percent changes in anti-inflammatory (IL-10) and the pro-inflammatory (TNF-α) cytokines in the co-culture medium are shown for both groups. The graph shows that all of the samples from the sarcoidosis subjects shift towards an anti-inflammatory state (increase in IL-10 and decrease in TNF-α), while samples from the control subjects do not (*n* = 9 for sarcoidosis subjects, *n* = 7 for control subjects, unpaired student’s *t* test; data are shown as mean ± SEM; *p* = 0.011 for IL-10 and 0.029 for TNF-α).

**Figure 2 jcm-09-00278-f002:**
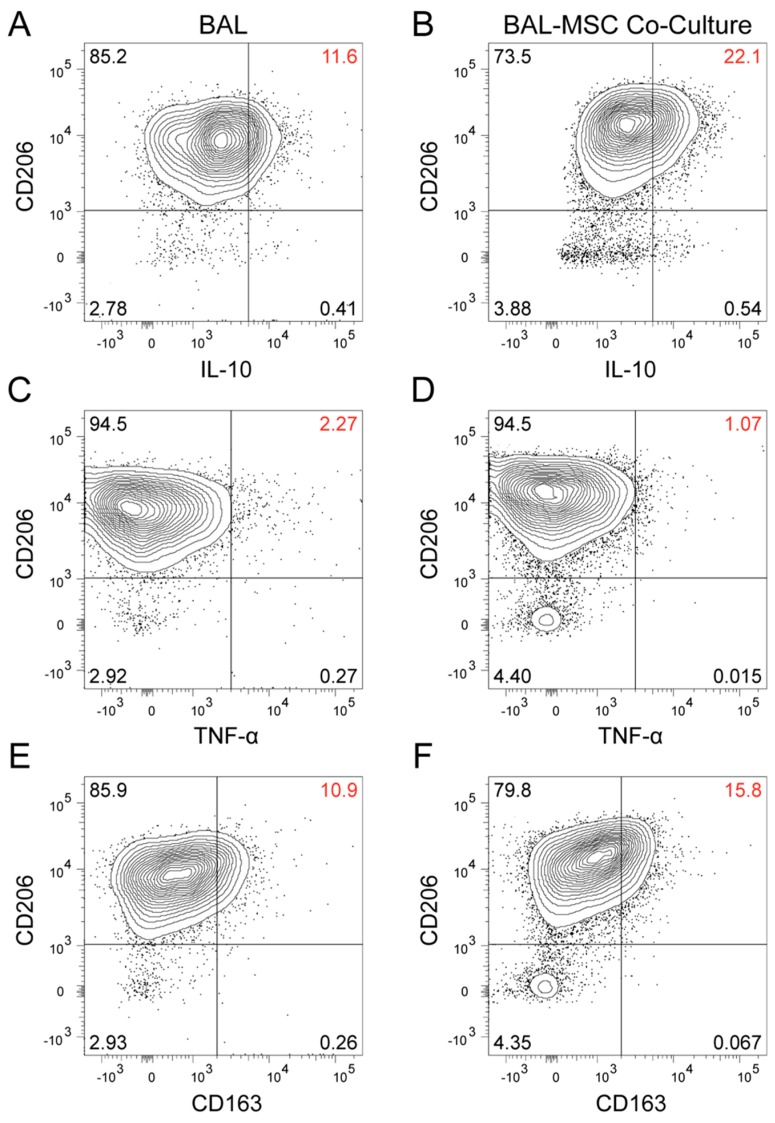
Flow cytometry analysis of mononuclear cells (90% of which were macrophages) freshly prepared from a sarcoidosis subject (sarcoidosis subject 16). Bronchoalveolar lavage (BAL) cells were plated and cultured for 16 h with (**B**,**D**,**F**) and without (**A**,**C**,**E**) the presence of bone marrow stromal cells (MSCs). Surface marker CD206 was used to identify alveolar macrophages (AMs), and intracellular flow cytometry was performed to determine changes in their cytokine production. Co-cultured BAL cells increased their IL-10 (**A**,**B**) and decreased their TNF-α (**C**,**D**) production. CD163 (a surface marker thought to indicate an anti-inflammatory state) also increased (**E**,**F**). These changes suggest a shift from a pro-inflammatory towards an anti-inflammatory state of AMs following contact with MSCs.

**Figure 3 jcm-09-00278-f003:**
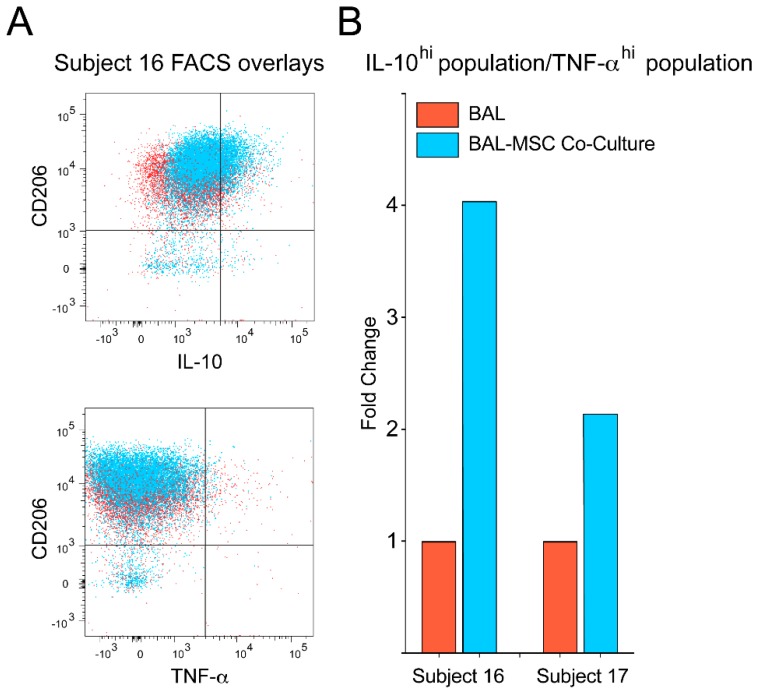
Intracellular cytokine ratios of alveolar macrophages (AMs) suggest a change from M1 (pro-inflammatory) to M2 (anti-inflammatory) status after co-culture with bone marrow stromal cells (MSCs). (**A**) Bronchoalveolar lavage (BAL) cells are depicted in red, BAL cells from co-cultures are pictured in blue. The co-cultured BAL cells demonstrate a clear shift towards the right, indicating an increase in the mean fluorescent intensity (MFI) of the IL-10+ population, and upwards, indicating an increase in the MFI of CD206 in these cells. Due to the lower percentage of cells producing TNF-α, a leftward shift is not as evident, although an upward shift indicating increased MFI of CD206 in these cells is present. (**B**) The fold change of the ratio of IL-10 to TNF-α for the BAL–MSC co-culture is shown for both subjects (sarcoidosis subjects 16 and 17). BAL cells cultured alone are depicted by the red bars (the ratio was converted to a value of 1), and co-cultured BAL cells are depicted by the blue bars (representing the fold increase of the ratio). The bar graph shows that—although not to the same extent—the BAL cells shifted towards a more anti-inflammatory state in both subjects.

**Figure 4 jcm-09-00278-f004:**
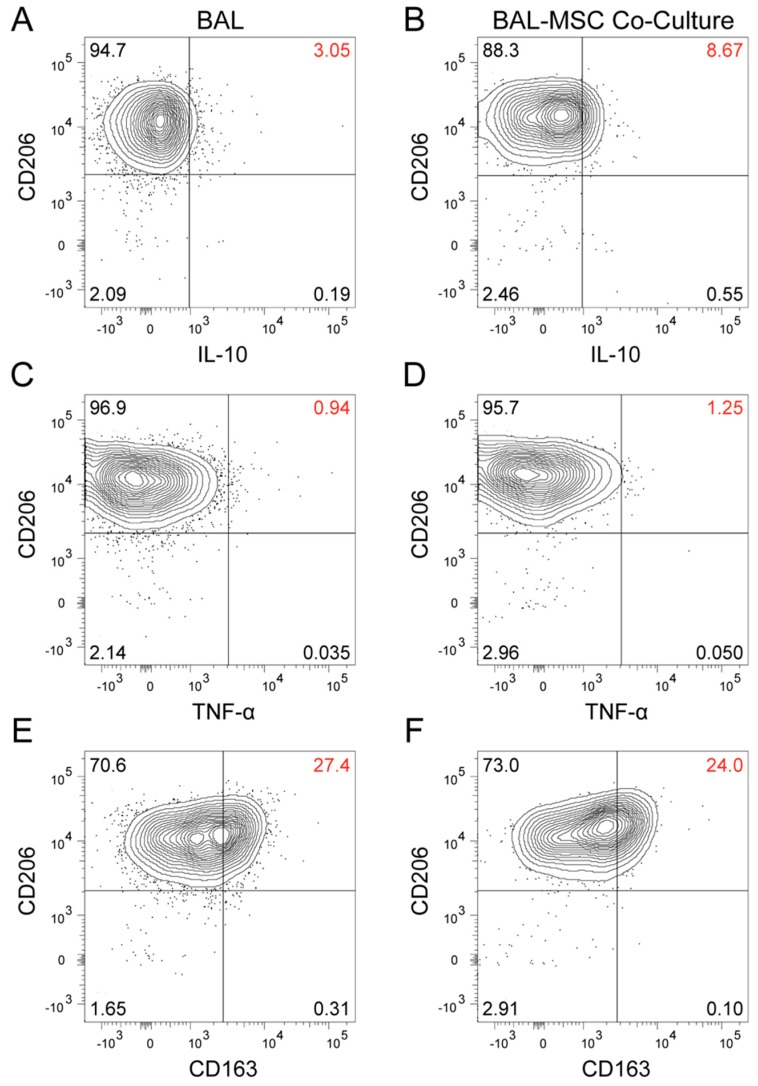
Flow cytometry analysis of mononuclear cells (73% of which were macrophages) freshly prepared from a sarcoidosis subject (sarcoidosis subject 17). Bronchoalveolar lavage (BAL) cells were plated and cultured for 16 hours with (**B**,**D**,**F**) and without (**A**,**C**,**E**) the presence of bone marrow stromal cells (MSCs). Surface marker CD206 was used to identify alveolar macrophages (AMs), and intracellular flow cytometry was performed to determine changes in their cytokine production. Co-cultured BAL cells almost tripled their IL-10 (**A**,**B**) while their TNF-α (**C**,**D**) production barely changed. Interestingly, CD163 (a surface marker thought to indicate an anti-inflammatory state) decreased (**E**,**F**). Overall, these changes may suggest a shift from a pro-inflammatory towards an anti-inflammatory state of AMs following contact with MSCs.

**Table 1 jcm-09-00278-t001:** Demographics and Basic Clinical Features.

Parameter	Controls (*n* = 7)	Sarcoidosis (*n* = 15)	*p*-Value
	**Mean Value**	**Mean Value**	
**Total Participants**			
**Female (number)/(%)**	4/(50%)	11/(73.3%)	NS
**Age (years)**	45.4	53.1	NS
**Race**			
**Black (number)/(%)**	5/(62.5)	8/(53.3)	NS
**White (number)/(%)**	2/(25)	5/(33.3)	NS
**Asian (number)/(%)**	1/(12.5)	0	NS
**Multiracial (number)/(%)**	0	2/(13.3)	NS
**Height (cm), (SD)**	174, (0.91)	168, (0.36)	NS
**Modification of MRC Dyspnea Scale**	0	0.9, (1.36)	0.027
**Inhaled Steroid (number/(%)), (SD)**	0	5/(33.3%), (0.49)	0.019
**Prednisone (number / (%)), (SD)**	0	2/(20%), (0.35)	NS

Abbreviations: Medical Research Council (MRC). SD: standard deviation. NS: not significant.

**Table 2 jcm-09-00278-t002:** Bronchoalveolar Lavage Cell Counts.

Parameter	Controls (*n* = 7)		Sarcoidosis (*n* = 15)		*p*-Value
	**Mean Value**	**SD**	**Mean Value**	**SD**	
**BAL Cell count (× 10^7^)**	21	9.7	22.6	14.8	NS
**BAL Lymphocytes (%)**	6.57	6.4	16.96	12.5	0.02
**BAL Macrophage (%)**	80.25	33.2	70.21	23.7	NS
**BALF % return (%)**	48.6	8.2	51.67	12.2	NS
**Viability of cells (%)**	80.57	15.7	84.6	8.0	NS

Abbreviation: NS: not significant.

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
