# Peer review of "Bone Marrow-Derived Mesenchymal Stromal Cells (MSCs) Modulate the Inflammatory Character of Alveolar Macrophages from Sarcoidosis Patients"

_jcm, 2020, doi:10.3390/jcm9010278_

Round 1
Reviewer 1 Report
Dear authors, this interesting report provides important pre-clinical data to motivate the treatment of sarcoidosis with intra-pulmonary allogeneic BM-MSC delivery. The paper is generally well written, the topic well introduced and discussed, and the patient stratification and clinical data appear to be sound. I only have some minor comments:
Minor comments:
1) Line 185-191 and Figure 1A+B: In Figure 1A, it is not entirely clear from the figure and corresponding legend what the arrow and the star is supposed to indicate. Please find a better description how these two patients diverged from the rest, maybe also include the patient numbers at the bottom of the graph so that it is easier to relate who is who? In Figure 1B, please also indicate the significant difference in the figure not only in the figure legend (e.g. P<0.01, typically indicated by three stars ***). Considering Figure 1B, interestingly, the IL10/TNFa-ratio was not at all affected/shifted upon coculture of healthy donor derived BAL/AMs with the same bone marrow MSCs, as contrasted to BAL/AMs from sarcoidosis subjects. The potentially underlying differences in cellular signaling should be discussed in detail, e.g. in the paragraphs around line 285-317.
2) Line 50-59 and 276-284: These mechanisms of macrophage polarization by MSCs are also nicely summarized in two recent reviews, PMID: 31214185 and 31214172. Please consider including these references in this paragraph, and please discuss if treatment efficacy and mechanism of action (MoA) would vary in response to intravascular vs. intrapulmonary MSC delivery (31417542 and 30711482) and what do you think would be the impact the using fresh vs. frozen MSCs (e.g. PMID: 27837556) since it appears that the authors used both in their study and this is also a very likely practical question in clinical application of the cells (please see line 93-115)???
3) Line 48-49 and 329-331 and 322-335: “In clinical trials to date, MSC have demonstrated an excellent safety profile” and “Phase 1 clinical trials using MSCs from bone marrow, placenta, and adipose tissue… have demonstrated a safe profile…”. Although I agree in principle to these two statements, both appear to generalize although there has been a large diversification in MSC products and delivery methods in the past 10 years and the authors should take into account that both the efficacy and safety profile of different types of MSCs strongly depends on the delivery method chosen for specific indications. The authors should briefly outline the difference between intravascular (for sarcoidosis intravenous would be most likely) vs. intratracheal / intrapulmonary delivery, since this may have very different effects considering efficacy and safety of the treatment (e.g. instant-blood mediated inflammatory reaction (IBMIR). For two very recent references that extensively discussed these issues please see PMID: 31417542 and 30711482. The approach of intrapulmonary delivery would be indeed very interesting to overcome any efficacy/safety concerns associated with IBMIR.
Author Response
First of all we would like to thank the reviewers for their comments and suggestions. We feel that the paper has been significantly improved by following their suggestions. Below, please, find our point by point responses and description of the changes we made.
REVIEWER 1
Dear authors, this interesting report provides important pre-clinical data to motivate the treatment of sarcoidosis with intra-pulmonary allogeneic BM-MSC delivery. The paper is generally well written, the topic well introduced and discussed, and the patient stratification and clinical data appear to be sound. I only have some minor comments:
Minor comments:
1) Line 185-191 and Figure 1A+B: In Figure 1A, it is not entirely clear from the figure and corresponding legend what the arrow and the star is supposed to indicate. Please find a better description how these two patients diverged from the rest, maybe also include the patient numbers at the bottom of the graph so that it is easier to relate who is who? In Figure 1B, please also indicate the significant difference in the figure not only in the figure legend (e.g. P<0.01, typically indicated by three stars ***). Considering Figure 1B, interestingly, the IL10/TNFa-ratio was not at all affected/shifted upon coculture of healthy donor derived BAL/AMs with the same bone marrow MSCs, as contrasted to BAL/AMs from sarcoidosis subjects. The potentially underlying differences in cellular signaling should be discussed in detail, e.g. in the paragraphs around line 285-317.
As you will see we have redesigned Fig.1 to make it easier for the readers to follow. As suggested, we added numbers to the waterfall graph to enable identification of the patients. We also separated the B panel into IL-10 vs. TNF-a to show that the difference is statistically significant in the decrease of the TNF-a as well as the increase of IL-10 production by the patients alveolar macrophages when compared to the volunteers. We describe in the Fig. legend the two patients who did not follow the trend (and we don't call them outliers) and we eliminated the asterisk and labelled these patients by framing their numbers in the bottom of the graph. All patients are included in the statistics shown in B. We added a paragraph (second paragraph of the Discussion) to explain why we think there is a stronger response of the AMs to MSCs when they both are in an inflammatory environment.
2) Line 50-59 and 276-284: These mechanisms of macrophage polarization by MSCs are also nicely summarized in two recent reviews, PMID: 31214185 and 31214172. Please consider including these references in this paragraph, and please discuss if treatment efficacy and mechanism of action (MoA) would vary in response to intravascular vs. intrapulmonary MSC delivery (31417542 and 30711482) and what do you think would be the impact the using fresh vs. frozen MSCs (e.g. PMID: 27837556) since it appears that the authors used both in their study and this is also a very likely practical question in clinical application of the cells (please see line 93-115)???
3) Line 48-49 and 329-331 and 322-335: “In clinical trials to date, MSC have demonstrated an excellent safety profile” and “Phase 1 clinical trials using MSCs from bone marrow, placenta, and adipose tissue… have demonstrated a safe profile…”. Although I agree in principle to these two statements, both appear to generalize although there has been a large diversification in MSC products and delivery methods in the past 10 years and the authors should take into account that both the efficacy and safety profile of different types of MSCs strongly depends on the delivery method chosen for specific indications. The authors should briefly outline the difference between intravascular (for sarcoidosis intravenous would be most likely) vs. intratracheal / intrapulmonary delivery, since this may have very different effects considering efficacy and safety of the treatment (e.g. instant-blood mediated inflammatory reaction (IBMIR). For two very recent references that extensively discussed these issues please see PMID: 31417542 and 30711482. The approach of intrapulmonary delivery would be indeed very interesting to overcome any efficacy/safety concerns associated with IBMIR.
We added another paragraph (towards the end of the Discussion) where we included the suggested references and addressed the issue of delivery routes, including discussion of IBMIR. We also added a few sentences about fresh vs. frozen cells.
Reviewer 2 Report
I have reviewed the manuscript entitled “MSCs modulate the inflammatory character of 2 alveolar macrophages from sarcoidosis patients” by Mc Clain-Caldwell.
The authors hypothesized that MSC cellular therapy, as a local treatment through the airways, might decrease lung inflammation in patients with sarcoidosis. Due to the lack of an appropriate animal model, they tested this hypothesis by isolating primary cells from sarcoid subjects’ BAL and cultured briefly in the presence and absence of healthy human MSCs. The authors collected BALs from consenting human subjects with pulmonary sarcoidosis (N=15) and normal volunteer controls (N=8) and examined the effect of bone marrow MSC (pooled from two random donors) co-culture on the immune cell physiology of culture adapted LPS-activated lung macrophages. The authors conclude that MSCs convert sarcoid BAL macrophages from a M1 to a M2 functionality as extrapolated from reduced TNFalpha and increased IL-10 production.
The strength if this report lies in their TNFalpha and IL-10 analysis of culture adapted primary sarcoid BAL macrophages derived from human subjects (N=15) and is commendable.
The impact of the observations made is blunted by the solely descriptive nature of investigation and mechanism(s) by which MSC interact with macrophages is implied in discussion but not tested or investigated.
The main weakness of this report is methodological.
In figure 1 legend the authors state that “TNFalpha and IL-10 concentrations were measured from tissue culture medium of LPS stimulated co-cultures of MSCs and BAL cells from sarcoidosis and control subjects. Of the 11 cases presented, 2 sarcoidosis samples diverged from the trend in one cytokine or the other (indicated by arrows and asterisks)… The two outlier sarcoidosis subjects are not plotted.” The authors go on to analyze data excluding censored samples from the sarcoidosis subjects and demonstrate a shift towards an anti-inflammatory state while the control subjects do not (n = 9 for sarcoidosis subjects, n = 7 for 203 control subjects, unpaired student’s t test; data are shown as mean ± SEM; p = 0.0079). The authors do not provide any rationale for censoring the data derived from these two subjects other than they were “outliers”. The authors also state in their discussion that this “outlier” response was also observed in a third subject whose cells served for flow analysis and not included in the 9 analyzed samples in Figure 1.
The authors’ censoring of samples included in analysis does not follow any rationale other than they were “outliers”. The concern is that inclusion of these two censored subjects in analysis provided would affect the power of this study to demonstrate a difference in the ratios between the anti-inflammatory (IL-10) and the pro-inflammatory (TNF-alpha) cytokines in the co-culture medium (Figure 1). If statistical significance were lost, then the conclusions drawn in title/abstract and discussion would be substantially affected. The authors need to address this explicitly.
Author Response
First of all we would like to thank the reviewers for their comments and suggestions. We feel that the paper has been significantly improved by following their suggestions. Below, please, find our point by point responses and description of the changes we made.
REVIEWER 2
I have reviewed the manuscript entitled “MSCs modulate the inflammatory character of 2 alveolar macrophages from sarcoidosis patients” by Mc Clain-Caldwell.
The authors hypothesized that MSC cellular therapy, as a local treatment through the airways, might decrease lung inflammation in patients with sarcoidosis. Due to the lack of an appropriate animal model, they tested this hypothesis by isolating primary cells from sarcoid subjects’ BAL and cultured briefly in the presence and absence of healthy human MSCs. The authors collected BALs from consenting human subjects with pulmonary sarcoidosis (N=15) and normal volunteer controls (N=8) and examined the effect of bone marrow MSC (pooled from two random donors) co-culture on the immune cell physiology of culture adapted LPS-activated lung macrophages. The authors conclude that MSCs convert sarcoid BAL macrophages from a M1 to a M2 functionality as extrapolated from reduced TNFalpha and increased IL-10 production.
The strength if this report lies in their TNFalpha and IL-10 analysis of culture adapted primary sarcoid BAL macrophages derived from human subjects (N=15) and is commendable.
The impact of the observations made is blunted by the solely descriptive nature of investigation and mechanism(s) by which MSC interact with macrophages is implied in discussion but not tested or investigated.
We published a very detailed study in Nature Medicine 10 years ago focused on the mechanism how MSCs work in an inflammatory environment to reprogram pro-inflammatory macrophages towards an anti-inflammatory phenotype (Nemeth, K.; Leelahavanichkul, A.; Yuen, P.S.; Mayer, B.; Parmelee, A.; Doi, K.; Robey, P.G.; Leelahavanichkul, K.; Koller, B.H.; Brown, J.M., et al. Bone marrow stromal cells attenuate sepsis via prostaglandin E(2)-dependent reprogramming of host macrophages to increase their interleukin-10 production. Nat Med 2009,15, 42-49). Recently, we also published two papers (PMID:30389270, PMID:30595353) confirming those findings and further connecting the immunosupressive behavior of MSCs to changes in IL-10 and TNFa in the macrophages that are in cell to cell contact with them. We built our present study on those earlier observations - thus did not feel the need to study the mechanism of the effect again.
The main weakness of this report is methodological.
In figure 1 legend the authors state that “TNFalpha and IL-10 concentrations were measured from tissue culture medium of LPS stimulated co-cultures of MSCs and BAL cells from sarcoidosis and control subjects. Of the 11 cases presented, 2 sarcoidosis samples diverged from the trend in one cytokine or the other (indicated by arrows and asterisks)… The two outlier sarcoidosis subjects are not plotted.” The authors go on to analyze data excluding censored samples from the sarcoidosis subjects and demonstrate a shift towards an anti-inflammatory state while the control subjects do not (n = 9 for sarcoidosis subjects, n = 7 for 203 control subjects, unpaired student’s t test; data are shown as mean ± SEM; p = 0.0079). The authors do not provide any rationale for censoring the data derived from these two subjects other than they were “outliers”. The authors also state in their discussion that this “outlier” response was also observed in a third subject whose cells served for flow analysis and not included in the 9 analyzed samples in Figure 1.
The authors’ censoring of samples included in analysis does not follow any rationale other than they were “outliers”. The concern is that inclusion of these two censored subjects in analysis provided would affect the power of this study to demonstrate a difference in the ratios between the anti-inflammatory (IL-10) and the pro-inflammatory (TNF-alpha) cytokines in the co-culture medium (Figure 1). If statistical significance were lost, then the conclusions drawn in title/abstract and discussion would be substantially affected. The authors need to address this explicitly.
Reviewer one also raised this valid point and here is our response:
We have redesigned Fig.1 to make it easier for the readers to follow. As suggested, we added numbers to the waterfall graph to enable identification of the patients. We also separated the B panel into IL-10 vs. TNF-a to show that the difference is statistically significant in the decrease of the TNF-a as well as the increase of IL-10 production by the patients alveolar macrophages when compared to the volunteers. We describe in the Fig. legend the two patients who did not follow the trend (and we don't call them outliers) and we eliminated the asterisk and labelled these patients by framing their numbers in the bottom of the graph. All patients are included in the statistics shown in B.
In the text we also explained the reason how these patients were different from the others and included both in the statistical evaluation and explain in the Discussion: " In our cytokine ELISA assays of co-culture medium, two sarcoidosis subjects did not follow the trend of decrease in TNF- a and increase in IL-10 (like all the other patients). In one subject (19 in Fig 1), both TNF- a and IL-10 decreased in the culture medium when his/her BAL cells were co-cultured with MSCs in the presence of LPS. This patient had an active sinus infection at the time of BAL harvest that might account for the deviation. The other unusual response had a slight increase in TNF-alpha, but a larger increase in IL-10 production (25 in Fig.1). This patient failed to stop his/her steroid intake as all the others did. It is important to mention though, that in spite of the deviation from the other responses, even in these two patients the change induced in the AMs by the MSCs were tilted towards anti-inflammatory change shifting the ratio towards less TNF- a and more IL-10. In a separate assay, one of the two sarcoidosis subjects whose BAL was analyzed by flow cytometry showed a similar change – a small increase in TNF-a and a larger increase in IL-10. This patient had clinically inactive disease overall."
Round 2
Reviewer 2 Report
my concerns have been sufficiently addressed